# Forest cover mediates large and medium-sized mammal occurrence in a critical link of the Mesoamerican Biological Corridor

Roberto Salom-Pérez[1,2,3]*, Daniel Corrales-Gutiérrez[1], Daniela Araya-Gamboa[1], Deiver Espinoza-Muñoz[1], Bryan Finegan[3], Lisanne S. Petracca[4]

**1** Panthera, New York, NY, United States of America, **2** Department of Fish and Wildlife Resources, University of Idaho, Moscow, Idaho, United States of America, **3** CATIE-Centro Agronómico Tropical de Investigación y Enseñanza, Turrialba, Costa Rica, **4** School of Aquatic and Fishery Sciences, University of Washington, Seattle, WA, United States of America

\* rsalom@panthera.org

**Data Availability Statement:** All main data are within the paper and its Supporting Information files. We have also ensured that our full data are

## Abstract

Connectivity of natural areas through biological corridors is essential for ecosystem resilience and biodiversity conservation. However, robust assessments of biodiversity in corridor areas are often hindered by logistical constraints and the statistical challenges of modeling data from multiple species. Herein, we used a hierarchical community occupancy model in a Bayesian framework to evaluate the status of medium and large-sized mammals in a critical link of the Mesoamerican Biological Corridor (MBC) in Costa Rica. We used camera traps deployed from 2013–2017 to detect 18 medium (1–15 kg) and 6 large (>15 kg) mammal species in a portion of two Jaguar Conservation Units (JCUs) and the Corridor linking them. Camera traps operated for 16,904 trap nights across 209 stations, covering an area of 880 km². Forest cover was the most important driver of medium and large-sized mammal habitat use, with forest specialists such as jaguars (*Panthera onca*) and pumas (*Puma concolor*) strongly associated with high forest cover, while habitat generalists such as coyotes (*Canis latrans*) and raccoons (*Procyon lotor*) were associated with low forest cover. Medium and large-sized mammal species richness was lower in the Corridor area ($\bar{x}$ = 9.78±1.84) than in the portions evaluated of the two JCUs ($\bar{x}$ = 11.50±1.52). Puma and jaguar habitat use probabilities were strongly correlated with large prey species richness (jaguar, $r$ = 0.59, p<0.001; puma, $r$ = 0.72, p<0.001), and correlated to a lesser extent with medium prey species richness (jaguar, $r$ = 0.36, p = 0.003; puma, $r$ = 0.23, p = 0.064). Low estimated jaguar habitat use probability in one JCU (Central Volcanic Cordillera: $\bar{x}$ = 0.15±0.11) suggests that this is not the jaguar stronghold previously assumed. In addition, the western half of the Corridor has low richness of large mammals, making it necessary to take urgent actions to secure habitat connectivity for mammal populations.

now freely available on the senior author's GitHub page (please see https://github.com/lisannepetracca/Salom_Perez_et_al_2021_PLOSOne).

**Funding:** RSP received support from Interamerican Development Bank (#CID/CCR/823/2013) https://www.iadb.org/, Small Cats Action Fund (#06-2016) https://www.panthera.org/grants-and-fellowships, and Kaplan Graduate Awards Program (#06-2017) https://www.panthera.org/grants-and-fellowships. The funders had no role in study design, data collection and analysis, decision to publish, or preparation of the manuscript.

**Competing interests:** The authors have declared that no competing interests exist.

## Introduction

Biodiversity is essential to maintain the resilience of ecosystems and the stability of their functions [1,2]. It is also critical for supporting a range of ecosystem services, reducing the risk of spread of infectious diseases, and maintaining productivity of several agricultural systems (e.g. facilitating pollination) [1,3,4]. While forests represent one of the most biodiverse ecosystems [5], global forest cover is being lost at a rate of 0.6% per year, largely due to conversion to agro-industrial land uses [6]. The increasing isolation of intact forest areas may not be enough to guarantee the long-term survival of species that require large spatial extents to sustain viable populations [7,8]. Thus, connectivity between separated forested areas through dispersal or biological corridors in human-dominated landscapes is crucial for the conservation of wide-ranging species [9,10].

With only 0.50% of the world's land area, the Mesoamerica region is recognized as a global hotspot, holding ~7% of the world's biodiversity [11]. Because of its relatively small size and its geographic position between North and South America, this isthmus has functioned for millennia as a natural bridge for wildlife, becoming arguably the most critical region for habitat connectivity in the Americas. Consequently, in 1997 the governments in the region created the Mesoamerican Biological Corridor (MBC), an initiative to preserve biodiversity and connect protected and other natural areas from southern Mexico to Panama [12]. Nevertheless, forest is being lost in several parts of the corridor, with approximately 271,600 ha lost across the region in the last ten years [13], and some researchers have already highlighted possible areas where connectivity may be broken or close to broken [9,14–17].

Medium and large-sized mammals play an important role in ecosystem dynamics, frequently representing the bulk of wildlife biomass in a given area [18], serving as predators exerting top-down control over other vertebrates [19], or performing ecological roles as prey species [20–22], seed consumers [23–25], or seed dispersers [24,26,27]. Medium and large-sized mammals are also among the most hunted animals by humans, and may comprise the main source of protein for some communities [18]. Importantly, large mammals incur substantial energetic costs and require large areas to maintain viable populations, placing many of them at higher risk of extinction and greater vulnerability to habitat alterations through human-related pressures [28–30].

The idea of large-sized mammals as "sentinel" species, decreasing in abundance or disappearing from areas where perturbations occur, has been a major driver of research into mammal ecology and distribution [29,31–33]. Much work has focused on single species under the "umbrella" concept, meaning that the conservation of one wide-ranging species will conserve other sympatric species [30,34–37]. However, some investigators suggest that the "umbrella" concept has not been tested appropriately [38–40], and single-species work is frequently confronted with small sample sizes that limit the analytical toolkit and scope of inference that can be applied to a specific area [41–43].

To overcome the problems of low sample sizes and/or detection rates, researchers often turn to occupancy models. Modern occupancy models account for imperfect detection by incorporating detection probability [43,44], producing less biased estimates compared to naive occupancy estimation [45]. The extension of occupancy models to a community or multi-species framework [46–48] allows the estimation of species and community-level occupancy by drawing species-level estimates from community-level hyperparameters [45,47–50], which can improve estimates for rare or elusive species. For these reasons, a multiple species approach to assess biodiversity, evaluate the effects of impacts or management actions, and evaluate connectivity has proven to be a useful alternative to the aforementioned methods [16,46–48,50].

In this paper, we use a multi-species community occupancy approach [46,50] to establish baseline information for medium and large-sized mammals, including two large predators, in

a critical wildlife corridor and the adjacent protected areas in Costa Rica. This corridor was primarily created to secure jaguar (*Panthera onca*; IUCN: near threatened; [51]) population connectivity. Thus, this area is considered a crucial corridor for the MBC and the Jaguar Corridor Initiative (JCI), the largest-scale carnivore conservation effort to date [9,15,52]. The JCI aims to preserve jaguar populations and range-wide habitat connectivity from Mexico to northern Argentina by identifying and securing dispersal corridors between core populations, also known as Jaguar Conservation Units or JCUs [9]. Jaguars, alongside pumas (*Puma concolor*; IUCN: least concern; [53]), are the largest predators in the region and their presence elsewhere has been associated with prey biomass and availability [22,54–56]. However, some studies have also found that prey richness can be related to large carnivore richness [57], occupancy [15], or dietary niche breadth [58].

Our objectives were to: (1) determine the environmental and human-related factors driving the occurrence of 24 medium ($N = 18$) and large ($N = 6$) native mammal species, plus the non-native domestic pig (an important prey item for jaguars and pumas in the study area), and (2) evaluate differences between the corridor and adjoining JCUs in terms of (a) species richness of medium and large-sized native mammal species and (b) habitat use probability for jaguars and pumas.

We expected that medium and large mammal species richness would be lower in the corridor than in the JCUs, given that the latter have higher forest cover and are classified as protected areas or indigenous territories. We also hypothesized that habitat use probability of jaguars and pumas would be related to large prey species richness given carnivores' high energetic requirements. We discuss the implications of our findings and the practicality of our model for this critical link of the MBC and the JCI, and for initiatives directed at preserving and monitoring biodiversity.

## Materials and methods

### Study area

The study area comprises the Barbilla-Destierro Biological Corridor (hereafter "the Corridor") and a portion of two adjacent JCUs: the Central Volcanic Cordillera (CVC) JCU and the Talamanca-Cordillera Central (TC) JCU (Fig 1). The JCUs are expert-defined areas that are believed to have resident jaguar populations, an adequate prey base, and high habitat quality for this species [8,59].

The CVC JCU comprises a continuous block of protected areas in central Costa Rica (IUCN categories II and VI; ~1,153 km$^2$) to the west of the Corridor, while the second JCU (TC JCU) is shared between Costa Rica and Panama and connects to the south-eastern side of the Corridor. The TC JCU encompasses a continuous group of protected areas (IUCN categories II and VI) and indigenous territories (~7,002 km$^2$) on the Costa Rican side. Primary and secondary forests represent ~75% of the JCU area, with the remaining area largely dominated by pasture [60].

The Corridor between those two JCUs is approximately 362 km$^2$, of which 58% is protected (IUCN category VI; Forest Reserves and a Protected Zone) or indigenous territory with a certain level of protection. No other suitable connections for jaguars have been identified between the CVC and TC JCUs, and more broadly between Nicaragua and Panama [15]. Almost all the area within the corridor is privately owned, comprising small farms. About 64% of the Corridor is covered by primary and secondary forest (2015 Aster Image; AIST https://gbank.gsj.jp/madas/), with the rest of the area dominated by pasturelands for livestock (20%), and agriculture (14%). There are two two-lane paved roads within the Corridor that connect the towns of Turrialba and Siquirres.

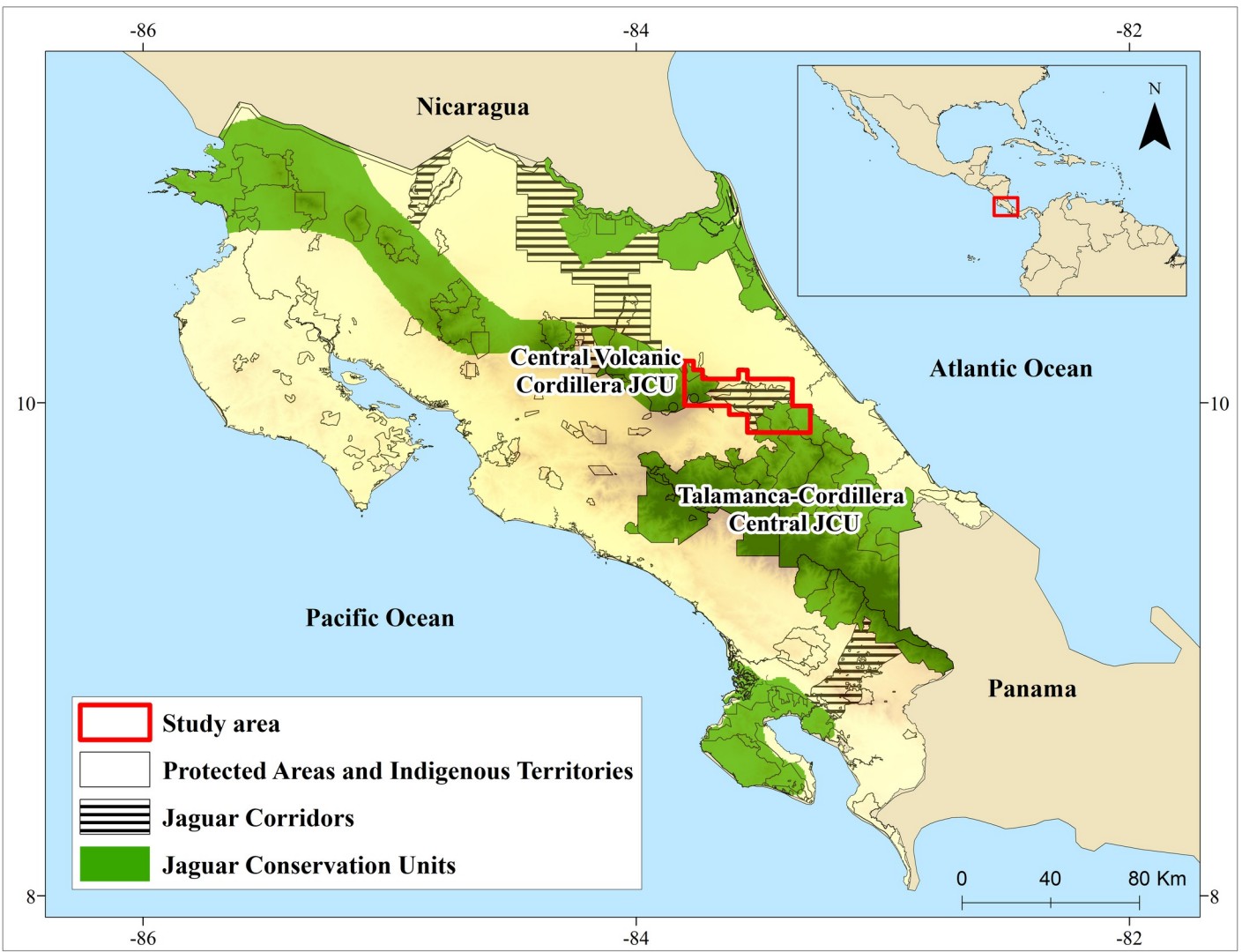

**Fig 1. Jaguar conservation units, jaguar corridors, and protected areas in Costa Rica.** Reprinted from Panthera and [60] under a CC BY license, with permission from Panthera and Repositorio TEC, original copyright 2013 and 2008.

## Study design and data collection

We conducted camera trap surveys to assess the presence of medium (1–15 kg) and large (>15 kg) mammal species over the study area. The study area was divided in four different blocks (CVC JCU, TC JCU, and Corridor Blocks 1 and 2). We created a grid system of 63 16 km$^2$ cells over the entire Corridor and portions of the CVC and TC JCUs (Fig 2). Cell size represented the approximate home range size of jaguars in Central America, one of the target species in our analyses, and presumably the species with the largest home range size [61–63].

The 16 km$^2$ grid cells were subdivided into four sub-cells of 4 km$^2$ each (Fig 3). We sampled two sub-cells per grid cell with two stations (one camera trap per station) in each sub-cell for a period of approximately three months. Selection of these sub-cells was random, but had to be adjusted when there was no forest, permits were not given, or access was difficult (e.g. presence of very steep slopes). Blocks of the study area were sampled at different times due to limitations in the number of cameras available and logistical considerations, as well as to allow for a higher

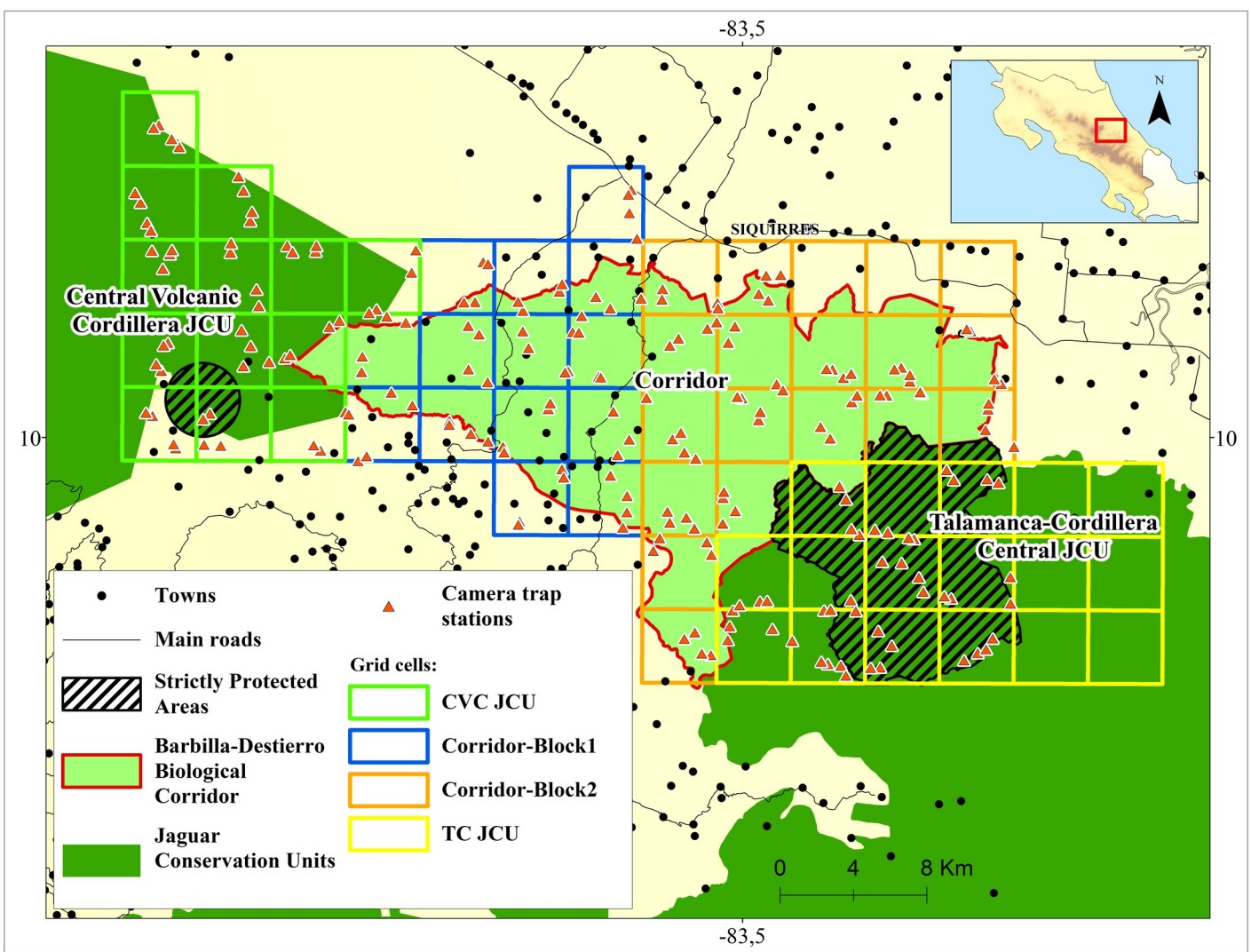

**Fig 2. Our sampling grid of 16 km² cells in the study area.** Camera traps to survey medium and large-sized mammals were placed in the CVC JCU (August 2014 to April 2015), Block 1 (Oct 2013—January 2014) and Block 2 (January 2014—May 2014) of the Corridor, and the TC JCU (September 2016 to April 2017). Study area: Barbilla-Destierro Biological Corridor (Corridor) and portions of the Central Volcanic Cordillera (CVC) and Talamanca-Cordillera Central (TC) Jaguar Conservation Units (JCUs). Strictly protected areas refer to IUCN Ia & II categories. Reprinted from Panthera and [60] under a CC BY license, with permission from Panthera and Repositorio TEC, original copyright 2013 and 2008.

density of cameras per area. Survey periods were Oct 2013—January 2014 (Corridor Block 1), January 2014—May 2014 (Corridor Block 2), August 2014 to April 2015 (CVC JCU), and September 2016 to April 2017 (TC JCU). No major land use or management changes occurred during these periods, though we account for potential baseline differences in mammal habitat use across blocks via random intercepts as described later.

We placed motion sensitive camera traps (Panthera® V3, V4, V5 and V6) in forested areas, strapped to trees at approximately 0.4–0.5 m parallel to the ground. Cameras were set to function continuously and to take three shots in every event during the day and one shot during the night. One camera of each 4 km² surveyed sub-cell was placed off a trail and the other one was placed on a human-made trail (when available), in an attempt to detect species that may avoid or use human trails (Fig 3).

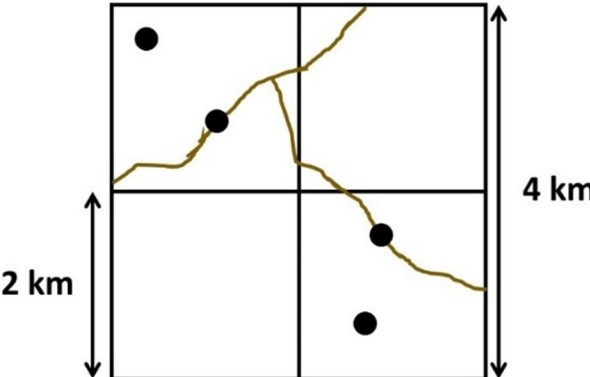

**Fig 3. Graphic representation of a 16 km² cell, 4 km² sub-cells and the planned distribution of the camera stations in the study area.** Cell: 16 km²; Sub-cells: 4 km². Black dots represent camera stations and brown lines represent human-made trails. Study area: Barbilla-Destierro Biological Corridor (Corridor) and portions of the Central Volcanic Cordillera (CVC) and Talamanca-Cordillera Central (TC) Jaguar Conservation Units (JCUs) from 2013–2017.

Camera location (±30 m) was recorded using a GPS device (Garmin®). Cameras were checked approximately every six weeks to download pictures and perform camera maintenance. Data were processed on Panthera IDS (Integrated Data Systems; version 1.13.786), where the species, date, time, and number of individuals in each photograph were recorded. For the occupancy-based analyses, we grouped data from all stations in each grid cell to make detection histories per week ("1" = detected, "0" = not detected, "NA" = inactive camera) for each of the 24 medium ($N = 18$ species) and large ($N = 6$) native mammal species, plus domestic pig (Total $N = 25$) (Table 1). We included domestic pig as they were frequently recorded and were consumed by jaguars and pumas in the study area, especially in indigenous areas where they roam freely in the forest (R. Salom-Pérez, *personal observation*).

Site covariates were selected *a priori* and hypothesized to be the main drivers of occupancy of medium and large sized-mammals in the study area [9,15,16,59,64,65] (S1 Table in S1 File). Site covariates expected to have a positive relationship with occupancy of medium and large mammals were enhanced vegetation index (EVI) [66], forest cover [65], distance to a primary road [65], distance from any human settlement, and distance from major human settlements (see detail in S1 File) [65]. Site covariates expected to have a negative relationship with occupancy of medium and large mammals were elevation [67], terrain ruggedness [68], distance to strictly protected areas (IUCN Ia & II categories) [65], distance to JCUs (Panthera unpub. data) and human presence (calculated as the number of human detections per 1,000 trap nights per stations; S1 File). All covariates were normalized prior to inclusion in the models such that the magnitude of regression coefficients could be compared within and among models [69]. To calculate effort, a covariate on detection, we calculated and standardized the sum of all trap nights on each occasion for every grid cell.

Permits for data collection were granted by the Costa Rican National System of Conservation Areas (SINAC; Spanish).

## Multi-level community occupancy model

Given that our survey occurred over a long period of time (~3.5 years), the closure assumption (i.e., there are no changes in the occupancy status of grid cells during the survey period [44]) is likely violated. Thus, occupancy ($\Psi$) is interpreted as probability of habitat use [70], and we assume that any changes in the occupancy status of grid cells over our survey period were random and that there were no major changes in the area throughout the study period.

**Table 1. Medium and large-sized mammal species included in the global analysis (N = 25).** Those included in the habitat use analysis for jaguars and pumas are indicated by a check mark. Barbilla-Destierro Biological Corridor and portions of the Central Volcanic Cordillera (CVC) and Talamanca-Cordillera Central (TC) Jaguar Conservation Units (JCUs), surveyed with camera traps from 2013–2017.

| Scientific name | Common name | Size | Included as prey for jaguars in the analysis | Included as prey for puma in the analysis |
|---|---|---|---|---|
| *Cabassous centralis* | Northern naked-tailed armadillo | medium | | |
| *Canis latrans* | Coyote | medium | | |
| *Conepatus semistriatus* | Skunk | medium | ✓ | ✓ |
| *Cuniculus paca* | Paca | medium | ✓ | ✓ |
| *Dasyprocta punctata* | Agouti | medium | ✓ | ✓ |
| *Dasypus novemcinctus* | Nine-banded armadillo | medium | ✓ | ✓ |
| *Didelphis marsupialis* | Opossum | medium | ✓ | ✓ |
| *Eira barbara* | Tayra | medium | | ✓ |
| *Galictis vittata* | Grison | | ✓ | |
| *Herpailurus yagouaroundi* | Jaguarundi | medium | | |
| *Nasua narica* | Coati | medium | ✓ | ✓ |
| *Leopardus pardalis* | Ocelot | medium | | |
| *Leopardus wiedii* | Margay | medium | | |
| *Leopardus tigrinus* | Oncilla | medium | | |
| *Procyon lotor* | Raccoon | medium | ✓ | ✓ |
| *Sylvilagus brasiliensis* | Rabbit | medium | ✓ | ✓ |
| *Tamandua mexicana* | Tamandua | medium | ✓ | ✓ |
| *Urocyon cinereoargenteus* | Gray fox | medium | | ✓ |
| *Mazama temama* | Red-brocket deer | large | ✓ | ✓ |
| *Odocoileus virginianus* | White-tailed deer | large | ✓ | ✓ |
| *Panthera onca* | Jaguar | large | | |
| *Pecari tajacu* | Collared peccary | large | ✓ | ✓ |
| *Puma concolor* | Puma | large | | |
| *Sus scrofa* | Domestic pig | large | ✓ | ✓ |
| *Tapirus bairdii* | Tapir | large | | |

We used a multi-level (or hierarchical) community occupancy model in a Bayesian framework to calculate jaguar and puma habitat use probability and species richness [46–48,50]. Unlike traditional occupancy models, the Bayesian framework can account for unobserved heterogeneity across species, space or time through random effects [15]. This type of heterogeneity could be expected in the current investigation given the relatively large spatial extent and the fact that the blocks were surveyed at different time periods.

To determine the covariates included in the global community occupancy model, we first ran single-species, single-season occupancy models for each medium and large-sized mammal species using R package RPresence [71]. We included effort as a covariate on detection in all models. On the occupancy side, we used all possible combinations of our 11 site covariates (additive only, no interactions), taking care to exclude variables from the same model if correlated at $|r| > 0.60$ ([72]; S2 Table in S1 File). In total, there were 79 models for each species (S2 File). Following Fieberg et al. [73], we then summed the log-likelihoods of each model across our 25 species and calculated Akaike Information Criterion with correction for small samples (AIC$_c$), with n = number of species [71]. This approach assumed independence between species observations [44].

Our final hierarchical occupancy model included forest cover, human presence, EVI, distance to strictly-protected area, and terrain ruggedness on habitat use, and effort on detection (S1 and S2 Tables in S1 File). Habitat use probability for each species at each site was estimated

as:

$$\text{Logit}\,(\Psi_{i,j}) = \xi_{il} + \alpha_i D_j \tag{1}$$

where $\xi_{il} \sim \text{Normal}(\mu_\xi, \tau_\xi)$ is the random intercept for each species $i$ at each block $l$ (CVC JCU, Corridor Blocks 1 and 2, and TC JCU), $\mu_\xi$ is the community-level or hyperparameter mean for the intercept on habitat use, and $\tau_\xi$ is its precision. The random intercept was used to account for potential spatial and temporal heterogeneity in habitat use due to differences by species and survey block [47,48,50]. In addition, $\alpha_i$ are estimated beta coefficients on habitat use for species $i$, where $\alpha_i \sim \text{Normal}(\mu\alpha, \tau\alpha)$; $\mu\alpha$ is the community-level or hyperparameter mean for each beta coefficient, and $\tau\alpha$ is its precision. $D_j$ are the standardized values for each covariate at grid cell $j$.

Detection probability for each species $i$ at each site $j$ and in each week $k$, was estimated as

$$\text{Logit}\,(p_{i,j,k}) = v_{il} + \beta_i effort_{j,k} \tag{2}$$

where $v_{il} \sim \text{Normal}(\mu_v, \tau_v)$ is the random intercept for each species $i$ at each block $l$, $\mu_v$ is the community-level mean for the intercept on detection, and $\tau_v$ is its precision. In addition, $\beta_i$ is the estimated beta coefficient for effort for species $i$, where $\beta_i \sim \text{Normal}(\mu_\beta, \tau_\beta)$, $\mu_\beta$ is the community-level mean for effort, $\tau_\beta$ is its precision, and $effort(j,k)$ are the standardized values for effort at grid cell $j$ on occasion $k$.

We defined true occurrence $z(_{i,j})$ as a binary variable in which $z(_{i,j}) = 1$ if species $i$ occurred in grid cell $j$ and = 0 otherwise. We modeled occurrence from a Bernoulli random variable, where $z_{i,j} \sim \text{Bern}(\Psi_{i,j})$, where $\Psi_{i,j}$ is the probability that species $i$ occurs at grid cell $j$. Importantly, species richness per grid cell was calculated as a derived parameter from the summation of $z_{i,j}$ values.

To account for imperfect detection, we modeled observed data $y(i,j,k)$ as $\text{Bern}(p_{i,j,k} * z_{i,j})$, where $p_{i,j,k}$ is the detection probability of species $i$ in grid cell $j$ in the survey occasion (week) $k$. The model accounted for the effect of species abundance on detection probabilities via species correlation parameter rho ($\rho$), a correlation between habitat use and detection probability [46].

We fit the Bayesian models in R 3.5.1 (R Core Team 2018) using package jagsUI [74], specifying three MCMC chains of 30,000 iterations, a burn-in of 5,000, and a thinning rate of three.

We estimated jaguar and puma habitat use probability as a function of species richness from the community model. We calculated species richness (again, via summation of $z_{i,j}$ values) for (1) all medium ($N = 18$) and large-sized ($N = 6$) native mammal species, including jaguar and puma and excluding the domestic pig; (2) jaguar and puma large prey species, including domestic pig ($N = 4$); (3) jaguar medium prey species ($N = 10$); and (4) puma medium prey species ($N = 11$ species). In order to select the medium and large prey species for jaguars and pumas, we conducted a literature review of publications on jaguar and puma diet and predation reports from Mexico to Panama (S4 Table in S3 File).

Lastly, we estimated the correlation between jaguar and puma habitat use probabilities and (1) medium prey richness and (2) large prey species richness, both of which were derived parameters from the prey community occupancy model as stated above.

## Results

Camera traps operated for 16,904 total trap nights across 209 stations. Fifty-five out of 63 total cells were surveyed, covering 87.30% of the study area (880 km$^2$). We registered 2,946 independent records ("independent" defined as records separated by 24 hours or occurring at different camera sites) of medium ($N = 18$) and large-sized ($N = 7$) mammal species, including domestic pig (S5 Table in S4 File).

The five most widespread species estimated by the Bayesian occupancy model were nine-banded armadillo (occurring in an estimated 89.32% of the area), coati (86.87%), ocelot (85.94%), tayra (82.35%) and jaguarundi (70.53%) (S5 Table in S4 File). Jaguars were less widespread in the study area (estimated to occur in 29.74% of the study area) compared to pumas (49.68%)

## Main environmental and human-related factors driving the presence of medium and large mammal species

The main driver of habitat use of medium and large-sized mammals was forest cover ($\alpha$ = 0.34; 95% Credible Interval (CRI) -0.21, 0.90), with lesser contributions of human presence ($\alpha$ = 0.10; 95% CRI -0.12, 0.31), EVI ($\alpha$ = -0.08; 95% CRI -0.57, 0.39), distance to strictly-protected area ($\alpha$ = -0.08; 95% CRI -0.47, 0.30), and terrain ruggedness ($\alpha$ = -0.02; 95% CRI -0.37, 0.32) (Table 2). Detection probability was positively associated with effort ($\beta$ = 0.39; 95% CRI 0.30, 0.49). There was a small positive correlation between habitat use and detection ($\rho$ = 0.26; 95% CI = -0.04 to 0.91).

All variables informing habitat use had a 95% CI overlapping zero, suggesting imprecision in parameter estimation and likely high variability of covariate effects among species (Table 2 and Fig 4 and S1-S4 Figs in S5 File) [46]. Percent forest had the greatest influence on species richness, with 89.26% of its posterior distribution above 0 and an effect size more than three times that of any other covariate (Table 2). At the species level, this relationship was clear (i.e., 95% CI did not overlap 0) and positive for collared peccary, jaguar, domestic pig, paca, puma, agouti and ocelot, while it was negative for coyote, nine-banded armadillo and raccoon (Fig 4). There was no clear effect of forest for the other species.

## Medium and large-sized mammal species richness in the Corridor and the two JCUs

Medium and large-sized mammal species richness estimates per grid cell ranged from 6 to 15 (95% CI 6–16; $\bar{x}$ = 10.60 ± SD 1.90) of 24 total species (domestic pig not included), with the TC JCU having the highest species richness estimate overall ($\bar{x}$ = 11.86 ± SD 1.15), followed by the CVC JCU ($\bar{x}$ = 11.03 ± SD 1.79), and Corridor ($\bar{x}$ = 9.78 ± SD 1.84) (Fig 5A).

## Jaguars, pumas and prey species

The JCUs had higher large prey species richness ($\bar{x}$ = 1.41 ± SD 0.58) and puma habitat use probability ($\bar{x}$ = 0.70 ± SD 0.20) compared to the corridor area (large prey richness: $\bar{x}$ = 0.59 ± SD 0.59; puma habitat use: $\bar{x}$ = 0.30 ± SD 0.21) (Figs 5B and 6b).

**Table 2. Community-level summary of hyperparameters for the covariates on detection and habitat use in the top model driving occurrence of medium and large wild mammals and domestic pig (n = 25).** We present the posterior means with standard deviation and 95% credible intervals, and an indicator of convergence ($\hat{R}$), for which values <1.1 indicate convergence. Barbilla-Destierro Biological Corridor and portions of the Central Volcanic Cordillera (CVC) and Talamanca-Cordillera Central (TC) Jaguar Conservation Units (JCUs), surveyed with camera traps from 2013–2017.

| Model parameter | Covariates | Beta (SD) | 95% credible interval | 50% credible interval | $\hat{R}$ |
|---|---|---|---|---|---|
| Habitat use ($\Psi$) | Percent forest cover (forest v2) | 0.34 (0.28) | -0.21, 0.91 | 0.16, 0.52 | 1.000 |
| | Human presence (human detections per 1,000 trap nights) | 0.10 (0.11) | -0.12, 0.31 | 0.03, 0.17 | 1.002 |
| | Mean Enhanced Vegetation Index | -0.08 (0.24) | -0.57, 0.39 | -0.23, 0.08 | 1.000 |
| | Mean distance to strictly-protected area | -0.08 (0.19) | -0.47, 0.30 | -0.20, 0.04 | 1.001 |
| | Mean Ruggedness | -0.02 (0.18) | -0.37, 0.32 | -0.13, 0.10 | 1.000 |
| Detection (p) | Effort (sum of all trap nights on each occasion) | 0.39 (0.05) | 0.30, 0.49 | 0.36, 0.42 | 1.000 |

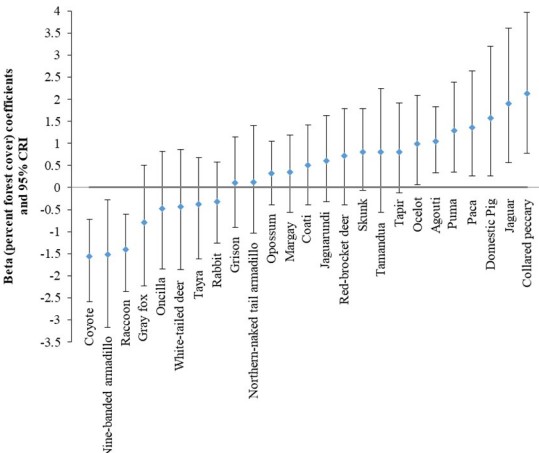

**Fig 4. Community-level hyperparameter estimates (with 95% Bayesian Credible Intervals) for the influence of forest cover on habitat use (Ψ) of medium and large mammals and domestic pig ($N$ = 25 species).** Barbilla-Destierro Biological Corridor (Corridor) and portions of Central Volcanic Cordillera (CVC) and Talamanca-Cordillera Central (TC) Jaguar Conservation Units (JCUs), surveyed with camera traps from 2013–2017.

On the other hand, jaguar and puma medium prey species richness was slightly higher in the eastern side of the Corridor (Block 2; jaguar medium prey (jmp): $\bar{x}$ = 6.14 ± SD 1.02; puma medium prey (pmp): $\bar{x}$ = 6.87 ± SD 0.94) in comparison to the rest of the Corridor (Block 1; jmp: $\bar{x}$ = 4.71 ± SD 1.13; pmp: $\bar{x}$ = 5.41 ± SD 1.02), the CVC JCU (jmp: $\bar{x}$ = 4.88 ± SD 1.16; pmp: $\bar{x}$ = 5.23 ± SD 1.01) and the TC JCU (jmp: $\bar{x}$ = 5.78 ± SD 0.79; pmp: $\bar{x}$ = 6.50 ± SD 0.7.0) (Fig 5C and 5D).

While the two JCUs performed similarly on most other metrics, they greatly differed with respect to jaguar habitat use, with jaguars having very low probability of habitat use in the CVC JCU ($\bar{x}$ = 0.15 ± SD 0.11) in comparison to TC JCU ($\bar{x}$ = 0.58 ± SD 0.16) (Fig 6A).

Estimates of jaguar and puma habitat use were strongly correlated with large prey species richness (jaguar, $r$ = 0.59, p<0.001; puma, $r$ = 0.72, p<0.001), and correlated to a lesser extent with medium prey species richness (jaguar, $r$ = 0.36, p = 0.003; puma, $r$ = 0.23, p = 0.064).

## Discussion

To our knowledge this is the most intensive camera trap study on a continuous area in Costa Rica [75], and the first to take place in this critical link between two JCUs. We found that forest cover was the main driver of medium and large-sized mammal habitat use at the community level, with evidence of widely differing relationships at the level of individual species. The hierarchical community approach allowed for the estimation of species and community-level effects, and for the incorporation of data from rare species. The model also accounted for heterogeneity in the sampling process through the incorporation of random effects.

While the medium and large-sized mammal species richness in the Corridor is slightly lower ($\bar{x}$ = 9.78±1.84 spp.) in comparison to the JCUs ($\bar{x}$ = 11.50±1.52), the number of records of large-sized species, is considerably lower in the former (Corridor: 38 independent records in 8,952 trap nights vs JCUs: 200 independent records in 7,952 trap nights). Additional research on the CVC JCU (where jaguar habitat use was low) is necessary to evaluate the viability of this critical link of the MBC and the JCI for jaguars, as jaguar records were entirely on the east side of the study area (Corridor Block 2 and TC JCU). This baseline information will be of paramount value to measure the outcome of conservation initiatives in the years to

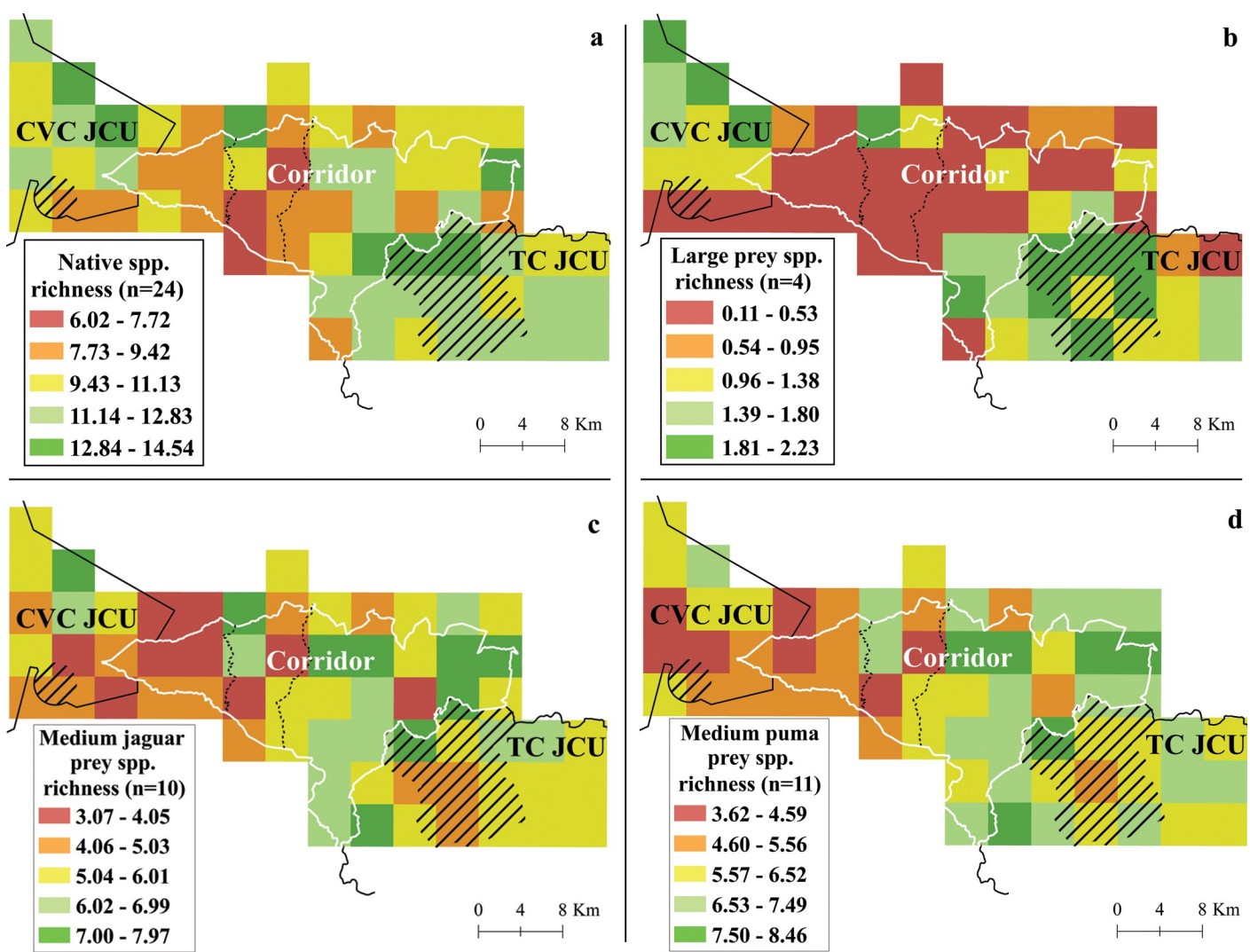

**Fig 5. Richness of native (a) medium and large-sized mammals, (b) large prey, (c) medium jaguar prey, and (d) medium puma prey.** The two main roads are shown in dotted lines; and the strictly protected areas are shown as striped polygons. Barbilla-Destierro Biological Corridor (Corridor, white polygon) and portions of Central Volcanic Cordillera (CVC) and Talamanca-Cordillera Central (TC) Jaguar Conservation Units (JCUs) (black outline), surveyed with camera traps from 2013–2017. Reprinted from Panthera and [60] under a CC BY license, with permission from Panthera and Repositorio TEC, original copyright 2013 and 2008.

come, especially related to the recently-constructed hydroelectric project in the middle of the corridor and related mitigation actions.

Forest cover was the most important covariate in our model related to medium and large-sized mammal habitat use, having an effect that was at least three times higher than any other covariate. This covariate seemed to be especially important for certain species, including the collared peccary, jaguar, domestic pig, puma, agouti and ocelot. These species, with exception of the domestic pig, are known to depend on, or at least be associated with, vegetation cover [51,53,76–78]. Almost all domestic pigs we detected were in the eastern side of the corridor, specifically in or near indigenous territories with high forest cover. These animals belong to the indigenous people and roam freely in the forest. On the other hand, coyote, nine-banded armadillo and raccoon seem to avoid areas with high forest cover in the study area. This was not surprising, as these are adaptable species that can be found in open and/or disturbed areas

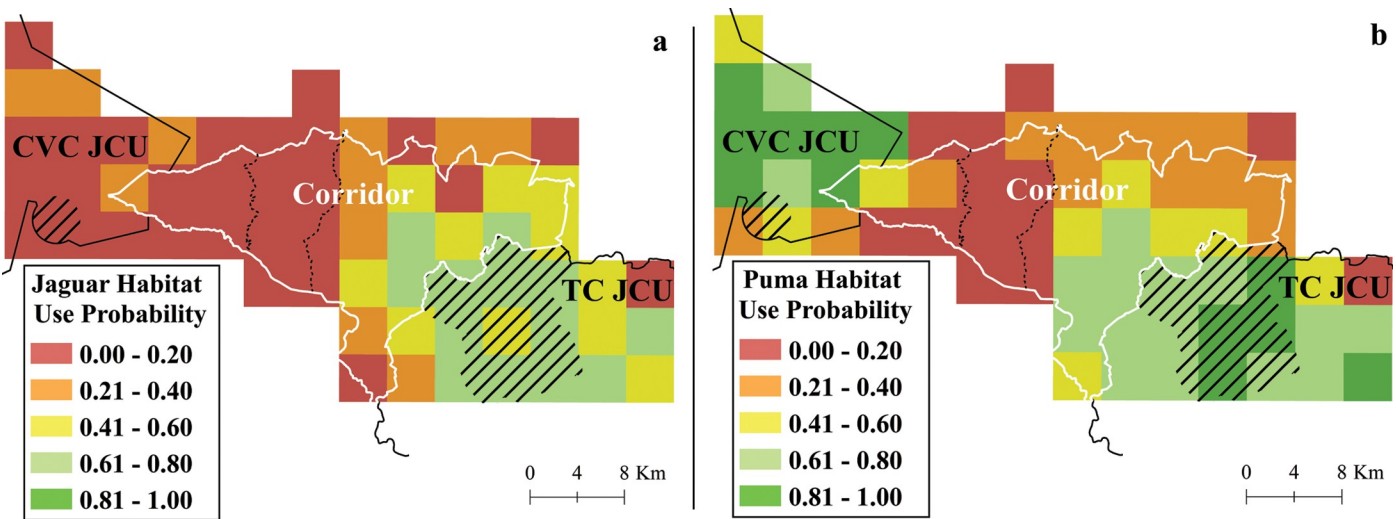

**Fig 6. Habitat use probability for (a) jaguar and (b) puma.** The two main roads are shown in dotted lines; and the strictly protected areas are shown as striped polygons. Barbilla-Destierro Biological Corridor (Corridor, white polygon) and portions of Central Volcanic Cordillera (CVC) and Talamanca-Cordillera Central (TC) Jaguar Conservation Units (JCUs) (black outline), surveyed with camera traps from 2013–2017. Reprinted from Panthera and [60] under a CC BY license, with permission from Panthera and Repositorio TEC, original copyright 2013 and 2008.

[76,79–81]. Thus, our model results were in line with disturbance-related species ecology and can help inform management decisions.

Contrary to our predictions, human presence and EVI were positively and negatively associated with overall species habitat use, respectively, though both effects were weak. While human presence is often used as a proxy for disturbance and has found to be negatively associated with habitat use at the community level [48], in our case the presence of humans may have a different interpretation. For example, people in our study area (e.g. tourists, hunters, indigenous people) may be actively looking for these animals and selectively walking on the same trails or in the same areas where they occur, leading to a positive association with mammal habitat use. This potentially important result requires additional study.

As we anticipated, medium and large-sized native mammal species richness was lower in the Corridor area than in the two JCUs. Nonetheless, these differences are subtle, indicating that there are some areas of the Corridor that are still in good condition, especially in the eastern portion close to the TC JCU. The lowest values for richness for these mammals, especially for large prey species, as well as for jaguar and puma habitat use probabilities, were in the western half of the corridor.

Low jaguar habitat use probability in the western JCU (CVC) was unexpected, given its apparent suitability for this species based on forest cover and presence of prey [52,65]. However, jaguar presence is likely not supported by other characteristics of the area, as most of the surveyed area in this JCU is not strictly protected (IUCN category VI), the terrain is very rugged and some areas have high elevation (2,000–3,300 m.a.s.l.) [51,82]. The highest habitat use probability for jaguars was located inside or near a strictly protected area, Barbilla National Park (IUCN category II) in the eastern JCU (TC). In contrast, pumas have high habitat use probability in both JCUs. Pumas are known to occur more frequently in higher elevations than jaguars, and could be benefitting from the apparent absence of a direct competitor in the CVC JCU [53,82,83]. A recent investigation further west into the CVC JCU found no sign of jaguar, adding extra support to the hypothesis that this area is not the jaguar stronghold previously assumed (Velado et al. unpub. data). It remains to be established whether low jaguar presence

in this JCU is explained by the site conditions mentioned above or if it is the result of more historical pressures (e.g. hunting, isolation).

Low richness of medium and large-sized prey within the Corridor was largely found between the two main roads. Although the proximity to road covariate did not make our global habitat use model, this and other types of infrastructure are known to have a negative effect on the presence of certain mammal species [32,84,85]. We detected the presence of large prey species, jaguars and pumas near these roads, but they were absent from the area between them. This area has less forest cover and higher settlement density, factors highly related to the presence of roads. Additionally, they are the Corridor grid cells located at greatest distance from the strictly protected areas and JCUs. We recommend further investigation into the potential barrier effect of these roads (and not just evaluate road proximity or density), such as the use of telemetry and non-invasive genetic tools to investigate movement and gene flow [14,84,86].

We found that puma and jaguar habitat use probabilities were strongly correlated with large prey species richness and, to a lesser degree, with medium prey species richness. Our results are consistent with other research on trophic interactions [57] and energetic constraints [87] showing a stronger link between large-bodied mammal predators and large mammal prey species in comparison to smaller prey species. The relationship between jaguar and puma occupancy with prey richness should be explored further in future studies, controlling for potentially confounding factors that can be correlated with prey richness, such as forest cover and biomass.

## Conclusions

Our results highlight the importance of generating on-the-ground information on the status of multiple species within population source sites and corridor areas, as well as using a hierarchical modeling framework for robust parameter estimation at the community and individual species levels to inform management decisions. The ability to account for heterogeneity in the sampling process (e.g. data taken over several sites and different years) makes this model versatile and easily adapted to different species and study systems.

Urgent actions are needed to secure connectivity of mammal populations in our study area and within the greater Mesoamerican Biological Corridor, and should focus on (1) increasing forest cover in the western half of the Corridor, notably between the two main roads, and (2) increasing habitat quality and conditions for prey species, with a particular emphasis on large species.

## Supporting information

**S1 File. Additional information on covariates selected a priori as being thought to have an influence on habitat use probability of medium and large mammals.** Barbilla-Destierro Biological Corridor (Corridor) and portions of Central Volcanic Cordillera (CVC) and Talamanca-Cordillera Central (TC) Jaguar Conservation Units (JCUs), surveyed with camera traps from 2013–2017.
(DOCX)

**S2 File. Individual species results and AICc values for the 79 models evaluated for medium and large-sized native mammals and domestic pig (n = 25).** Barbilla-Destierro Biological Corridor (Corridor) and portions of Central Volcanic Cordillera (CVC) and Talamanca-Cordillera Central (TC) Jaguar Conservation Units (JCUs), surveyed with camera traps from 2013–2017.
(XLSX)

**S3 File. Additional description on literature review conducted to select species included as prey species for jaguars (Panthera onca) and pumas (Puma concolor).** Barbilla-Destierro Biological Corridor (Corridor) and portions of Central Volcanic Cordillera (CVC) and Talamanca-Cordillera Central (TC) Jaguar Conservation Units (JCUs), surveyed with camera traps from 2013–2017.
(DOCX)

**S4 File. Additional information on number of occupied cells, relative abundance and number of independent detections of medium and large-sized native mammals and domestic pig (n = 25).** Barbilla-Destierro Biological Corridor (Corridor) and portions of Central Volcanic Cordillera (CVC) and Talamanca-Cordillera Central (TC) Jaguar Conservation Units (JCUs), surveyed with camera traps from 2013–2017.
(DOCX)

**S5 File. Species-level estimates for the influence of covariates on habitat use (Ψ) and detection (p) of medium and large-sized mammals.** Barbilla-Destierro Biological Corridor (Corridor) and portions of Central Volcanic Cordillera (CVC) and Talamanca-Cordillera Central (TC) Jaguar Conservation Units (JCUs), surveyed with camera traps from 2013–2017.
(DOCX)

## Acknowledgments

We thank Drs. Lisette Waits, Daniel Thornton, Fernando Casanoves, Ryan Long, Hugh Robinson, Nathaniel Robinson, Howard Quigley, and MSc. Carlomagno Soto for their valuable support and comments. We would like to thank the numerous technicians, researchers and volunteers that collaborated on the field work and data processing. We would like to give a special mention to the Panthera Costa Rica staff. We are most grateful to Barbilla-Destierro Biological Corridor local council, the Costa Rican Electricity Institute (ICE) and all property owners that collaborated with this project. We thank the Costa Rican National System of Conservation Areas (SINAC-MINAE) for providing the permits for this research. We also thank the Small Cats Action Fund, Kaplan Graduate Awards Program, Panthera, and the Inter-American Development Bank.

## Author Contributions

**Conceptualization:** Roberto Salom-Pérez, Daniela Araya-Gamboa.

**Data curation:** Roberto Salom-Pérez, Daniel Corrales-Gutiérrez, Deiver Espinoza-Muñoz, Lisanne S. Petracca.

**Formal analysis:** Roberto Salom-Pérez, Lisanne S. Petracca.

**Funding acquisition:** Roberto Salom-Pérez.

**Investigation:** Roberto Salom-Pérez, Daniel Corrales-Gutiérrez, Daniela Araya-Gamboa, Deiver Espinoza-Muñoz.

**Methodology:** Roberto Salom-Pérez, Lisanne S. Petracca.

**Project administration:** Roberto Salom-Pérez.

**Resources:** Roberto Salom-Pérez.

**Software:** Lisanne S. Petracca.

**Supervision:** Bryan Finegan, Lisanne S. Petracca.

**Validation:** Lisanne S. Petracca.

**Visualization:** Roberto Salom-Pérez.

**Writing – original draft:** Roberto Salom-Pérez.

**Writing – review & editing:** Roberto Salom-Pérez, Daniel Corrales-Gutiérrez, Daniela Araya-Gamboa, Deiver Espinoza-Muñoz, Bryan Finegan, Lisanne S. Petracca.

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
