## [Decision Letter · Decision Letter 0]

11 Sep 2020

PONE-D-20-25172

The status of medium and large-sized mammals in a critical link of the Mesoamerican Biological Corridor

PLOS ONE

Dear Dr. Salom-Perez,

Thank you for submitting your manuscript to PLOS ONE. After careful consideration, we feel that it has merit but does not fully meet PLOS ONE’s publication criteria as it currently stands. Therefore, we invite you to submit a revised version of the manuscript that addresses the points raised during the review process.

I have obtained reviews of this manuscript from two experts in the field. Reviewers concur that the manuscript is interesting and has some merits, however they all raise important and critical flaws that need to be addressed before the manuscript can be considered for publication. In particular, both reviewers have issues on the objectives and hypotheses framed, and that the interpretation of the effects of covariates is not correct. Reviewer 2 also raises concerns on the design, statistics and focus of the study, in particular on the relevance of assessing the corridor functioning, which is flagged as a key aim of the study. The reviewer also offered a number of very useful annotations directly on the manuscript.

We look forward to receiving your revised manuscript.

Kind regards,

Francesco Rovero, Ph.D.

Academic Editor

PLOS ONE

2. In your Methods section, please provide additional information regarding the permits you obtained for the work. Please ensure you have included the full name of the authority that approved the field sites access and, if no permits were required, a brief statement explaining why.

3. We note that Figures 1, 2, 6, 7 in your submission contain map images which may be copyrighted. All PLOS content is published under the Creative Commons Attribution License (CC BY 4.0), which means that the manuscript, images, and Supporting Information files will be freely available online, and any third party is permitted to access, download, copy, distribute, and use these materials in any way, even commercially, with proper attribution. For these reasons, we cannot publish previously copyrighted maps or satellite images created using proprietary data, such as Google software (Google Maps, Street View, and Earth). For more information, see our copyright guidelines: http://journals.plos.org/plosone/s/licenses-and-copyright.

3.1.    You may seek permission from the original copyright holder of Figures 1, 2, 6, 7 to publish the content specifically under the CC BY 4.0 license. 

3.2.    If you are unable to obtain permission from the original copyright holder to publish these figures under the CC BY 4.0 license or if the copyright holder’s requirements are incompatible with the CC BY 4.0 license, please either i) remove the figure or ii) supply a replacement figure that complies with the CC BY 4.0 license. Please check copyright information on all replacement figures and update the figure caption with source information. If applicable, please specify in the figure caption text when a figure is similar but not identical to the original image and is therefore for illustrative purposes only.

Reviewers' comments:

Reviewer's Responses to Questions

**Comments to the Author**

1. Is the manuscript technically sound, and do the data support the conclusions?

Reviewer #1: Yes

Reviewer #2: Partly

2. Has the statistical analysis been performed appropriately and rigorously? 

Reviewer #1: Yes

Reviewer #2: I Don't Know

3. Have the authors made all data underlying the findings in their manuscript fully available?

Reviewer #1: Yes

Reviewer #2: No

4. Is the manuscript presented in an intelligible fashion and written in standard English?

Reviewer #1: Yes

Reviewer #2: No

5. Review Comments to the Author

Reviewer #1: I found the present manuscript well written and presenting very valuable information that can contribute to the design and implementation of biological corridors.

Some observations I have are:

Authors are hypothesizing that presence of puma and jaguar is associated with large prey species richness. However, in the Introduction I did not observe a rationale that supports that hypothesis. Arguably, prey biomass is more important than richness to determine predator presence. For example, in Los Llanos of Venezuela, 70% of jaguar diet is made up of 4 species (2 peccaries, capybara and giant ant eater) (Polisar et al 2003); a similar case is observed by Aranda (1994) in Mexico and other studies. It will be helpful to expand the introduction to address this hypothesis.

Results of most covariates used to describe habitat use by predators and prey are non-conclusive. For example, in the case of human presence, distance to protected area and terrain ruggedness, 95% CRI overlap zero for all species. Only forest cover and EVI present conclusive results for some species (i.e., CRI do not overlap zero). As the first objective of this work is to “determine the main environmental and human-related factors driving the presence of 24 medium (N=18) and large (N=6) native mammal species…”, I suggest authors should test the effect of other predictors that might be of greater importance to determine species presence. For instance, authors mention the possible effect of roads on jaguar and puma. I think authors should include a measure of roads (e.g., distance to roads or density of roads) in their models, and remove covariates that seem to be not contributing to describe habitat use or species presence. Current work that evaluates the effect of roads on jaguar abundance can support that analysis, for example, see Espinosa et al. 2018: https://doi.org/10.1371/journal.pone.0189740.

I found the results of forest cover as a predictor of species presence very interesting; with a positive relationship with animals sensitive for habitat degradation, and a negative relationship with opportunistic species that “flourish” in an anthopogenized landscape. I think authors could put more emphasis on this finding and expand their discussion on this topic. Also, based in that finding I think authors should explore different models (i.e., different sets of covariates) for different species guilds or groups. For example, for small and less hunted species, habitat covariates, such as forest cover, proximity to water, terrain ruggedness among others might be more important than human-related covariates. However, for highly hunted species, such as peccaries, or species undergoing retaliatory killing, such as jaguar and puma, human-related covariates might be more important in determining their presence (e.g., human density, distance to road, distance to towns, etc.).

For the reader, it would be helpful to organize the manuscript in different subsections, for example, in methodology: Study area, wildlife surveys, Statistical analyses, etc. Also, Results and Discussion could be organized under the two hypothesis authors are testing: 1) species richness is higher in better conserved areas, 2) habitat use of large predators is related with large prey species richness.

Minor comments:

Line 124-125: “. This JCU…” not clear if it is the first or the second, clarify

Table 2. I think there is a mistake in a covariate name “Percent forest version 2”; correct in final version. Why is a 50% CRI being reported? I think there is no need for that.

Line 356: Consider not citing work “in preparation”; not a real reference.

Lines 385-386: This is a speculation that needs some justification; whether jaguar or pumas establish between roads depends on how far apart roads are, what is between roads and along them.

Reviewer #2: The paper is an assessment of the mammal fauna in this area (which happens to have a corridor), coupled with a more in-depth analysis of the underlying drivers affecting the occupancy of jaguars and pumas in the region. Being in a corridor (or not) does not seem to be a contrast to be estimated in the model although the authors do compare some of the results between the three sub-study areas (one in a corridor and two not). As such the paper seems unfocused covering several questions with little rigor and no insights into the corridor issue itself.

I have issues with the design of the study, in particular whether camera trap deployments were combined for a given cell or considered as independent measurement units within the cell. Also, camera traps were deliberately placed in trails, which can bias the measurement towards some species and increase the number of people captured. This is not ideal given that the assessment has a large geographical breadth and these placement decisions can affect the magnitude and precision of covariates at regional scales. The authors need to justify this design more clearly, explain how observations were aggregated for analysis, and explain what they mean by “independent observations”.

I was happy to see the use of modern statistical multi-species models being used in the analysis of these data and correcting for detection probability. However it was unclear how the authors arrived at their final model and how covariates were selected. I was disappointed that corridor/non-corridor was not a covariate in itself. The interpretation of some of the effects is incorrect (e.g. Fig 4) describing effects as positive or negative, when in reality they overlap with 0. The description of the statistical significance of some of the correlation coefficients is incorrect as well. When describing coefficients, effects or expected values the authors switch between mean+-confidence intervals and mean +- ? (Unclear wether it is standard error or standard deviation). For example when comparing the species richness between the three study areas (lines 312) it is unclear which parameter is used to describe uncertainty. If these are standard errors, there might not be any real difference between richness. If it is standard deviation, there might or might not be, the reader cannot assess it.

Both in the results and the discussion the authors seem to be more focused on the signs of the effects, then in their magnitude. Many of the effects discussed are not surprising (e.g. more forest more species). But there could be so much more here! In my opinion, the results —as presented— represent a lost opportunity since they might have important implications to better understand the effectiveness of corridors for mammals. As an example, if species richness within the corridor is not-significantly different compared to the more intact habitats being connected, this is an important result to be highlighted. Even if it is lower, the difference in mean species richness is only 1-2 species on average, which again is a very important result. These nuances in what the data is telling are key and often missed throughout the paper. The questions also seems too scattered, with some focusing on big cats, while others are looking at the whole community. It is confusing and difficult to get what the real results and messages of the research are.

Authors, please do not be discouraged by these comments. I congratulate you for this great effort. My comments are made in a constructive spirit, to bring out what I think could be an exceptional paper. I believe this research should be published, but the field design needs to be clarified, the questions need to be tightened and more focused, the process to reach the final model explicitly stated and the interpretation of the results put into a wider context. I have attached the MS with some additional comments and suggestions that I marked directly. The paper should also be reviewed by an English copy editor before resubmission to improve the style according to journal standards. Hope all this is useful.

6. PLOS authors have the option to publish the peer review history of their article (what does this mean?). If published, this will include your full peer review and any attached files.

Reviewer #1: No

Reviewer #2: No

---

## [Author Response · Author response to Decision Letter 0]

14 Jan 2021

Authors´Comment: We have carefully revised our manuscript completely according to the PLOS ONE style requirements. We made several changes to the first page (title, authors, affiliations). We´ve also included some changes in the format in the main body.

2. In your Methods section, please provide additional information regarding the permits you obtained for the work. Please ensure you have included the full name of the authority that approved the field sites access and, if no permits were required, a brief statement explaining why.

 Authors´Comment: We´ve included the information on the entity providing the permits in the Methods section.

3. We note that Figures 1, 2, 6, 7 in your submission contain map images which may be copyrighted. All PLOS content is published under the Creative Commons Attribution License (CC BY 4.0), which means that the manuscript, images, and Supporting Information files will be freely available online, and any third party is permitted to access, download, copy, distribute, and use these materials in any way, even commercially, with proper attribution. For these reasons, we cannot publish previously copyrighted maps or satellite images created using proprietary data, such as Google software (Google Maps, Street View, and Earth). For more information, see our copyright guidelines: http://journals.plos.org/plosone/s/licenses-and-copyright.

Authors´Comment: Figures 1, 2, 6 and 7 are the original work of the authors using original information, public layers from Panthera (the organization of the lead author) and the Costa Rican public Atlas (Ortiz-Malavasi E. Atlas digital de Costa Rica 2008. Cartago, Costa Rica; 2009). We are providing a permission letters from Panthera and TEC/Ortiz-Malavasi. 

 We have also ensured that our full data are now freely available on the senior author’s GitHub page (please see https://github.com/lisannepetracca/Salom_Perez_et_al_2021_PLOSOne).

 

Review Comments to the Author

Reviewer #1: I found the present manuscript well written and presenting very valuable information that can contribute to the design and implementation of biological corridors.

Some observations I have are:

Authors are hypothesizing that presence of puma and jaguar is associated with large prey species richness. However, in the Introduction I did not observe a rationale that supports that hypothesis. Arguably, prey biomass is more important than richness to determine predator presence. For example, in Los Llanos of Venezuela, 70% of jaguar diet is made up of 4 species (2 peccaries, capybara and giant ant eater) (Polisar et al 2003); a similar case is observed by Aranda (1994) in Mexico and other studies. It will be helpful to expand the introduction to address this hypothesis.

Authors’ comment: This is a great point by the reviewer, and we can certainly see the argument that prey biomass is a core component determining presence of predators. However, it has also been found that the richness of large-bodied prey can be a major driver of predator presence at large scales (please see Petracca et al. 2018 in Journal of Applied Ecology for an example including all jaguar corridors within Central America). It was the strong effect of this covariate in the Petracca et al. 2018 paper that led us to include this covariate in our manuscript - we have added material in our Introduction (L. 105-109) to help support our hypothesis that prey richness can be a viable driver of predator occupancy. 

Results of most covariates used to describe habitat use by predators and prey are non-conclusive. For example, in the case of human presence, distance to protected area and terrain ruggedness, 95% CRI overlap zero for all species. Only forest cover and EVI present conclusive results for some species (i.e., CRI do not overlap zero). As the first objective of this work is to “determine the main environmental and human-related factors driving the presence of 24 medium (N=18) and large (N=6) native mammal species…”, I suggest authors should test the effect of other predictors that might be of greater importance to determine species presence. For instance, authors mention the possible effect of roads on jaguar and puma. I think authors should include a measure of roads (e.g., distance to roads or density of roads) in their models, and remove covariates that seem to be not contributing to describe habitat use or species presence. Current work that evaluates the effect of roads on jaguar abundance can support that analysis, for example, see Espinosa et al. 2018: https://doi.org/10.1371/journal.pone.0189740.

Authors’ comment: Thank you for this valuable input. Given the feedback from both reviewers, it is obvious that we should have been much more clear about how our final Bayesian multi-species occupancy model was selected. We direct the reviewer to L. 209-221, where we have described the process of how we started with 11 covariates and used a maximum likelihood model selection approach to determine our final model. This protocol followed the statistical procedure of Petracca et al. 2018, mentioned above. Thus, we did indeed test for the explanatory value of other covariates, including distance to roads, but found them to be inconsequential compared to the final selected covariates. Please see Supplementary Material S1 & S2 for detailed information on our model selection process.

As to the observation that the effects of some covariates are inconclusive at the community level (and we expand upon this further in a response to Reviewer 2): this overlap of 95% CRIs with zero is common at this level due to (1) general uncertainty, given that certain species were detected very infrequently due to low detection probabilities, and (2) there is species-specific variation in parameter estimates. Examples of published papers with 95% community-level hyperparameters overlapping zero are Petracca et al. (2019) in Animal Conservation and Rich et al. (2016) in Journal of Applied Ecology. This was our motivation for providing 50% CRI estimates as well -- to provide a bit more nuance as to where the 25th to 75th percentile posterior densities were falling. We would also hesitate to remove certain covariates just because they are inconclusive at the community level, as their inclusion may prove important at the level of species-specific estimates. 

I found the results of forest cover as a predictor of species presence very interesting; with a positive relationship with animals sensitive for habitat degradation, and a negative relationship with opportunistic species that “flourish” in an anthopogenized landscape. I think authors could put more emphasis on this finding and expand their discussion on this topic. Also, based in that finding I think authors should explore different models (i.e., different sets of covariates) for different species guilds or groups. For example, for small and less hunted species, habitat covariates, such as forest cover, proximity to water, terrain ruggedness among others might be more important than human-related covariates. However, for highly hunted species, such as peccaries, or species undergoing retaliatory killing, such as jaguar and puma, human-related covariates might be more important in determining their presence (e.g., human density, distance to road, distance to towns, etc.).

Authors’ comment: We completely agree that we should have placed more emphasis on our finding that forest cover had different impacts depending on species’ sensitivity to habitat degradation. Please see L. 395-406 for how we have expanded upon this finding in the Discussion. 

As to your second point about delineating species guilds or groups, the big advantage of using a multi-species occupancy model is that the integration of data for all species allows for a “drawing of strength” such that species-specific parameters are estimated with greater precision (Dorazio & Royle 2005; Zipkin et al., 2009). This is particularly true for those species that were rarely photographed due to low detection probabilities. Thus, if we were to (post facto, which is likely problematic) divide the species into guilds and employ different covariates within each submodel -- that indeed goes against the very reason why we employed a multi-species model in the first place, and would almost certainly lead to less precision in our estimates and an inability to compare effect sizes across species. 

Lastly, the parameterization of our model still allows for the estimation of species-specific effects on occupancy. If covariates are not important for a given species, the species-specific estimates will simply have a posterior density around zero; if the covariate is indeed important, the estimates will have posterior densities that do not overlap zero (please see Figure 5 for an example of how much species-specific estimates ranged for forest cover). Thus, our model, as we have parameterized it, should not affect our ability to determine what covariates are important (or not) at the species level. 

For the reader, it would be helpful to organize the manuscript in different subsections, for example, in methodology: Study area, wildlife surveys, Statistical analyses, etc. Also, Results and Discussion could be organized under the two hypothesis authors are testing: 1) species richness is higher in better conserved areas, 2) habitat use of large predators is related with large prey species richness.

Authors’ comment: Great suggestion. We have improved our manuscript organization adding subsections in the Results section. 

Minor comments:

Line 124-125: “. This JCU…” not clear if it is the first or the second, clarify

Authors’ comment: We´ve changed the wording to specify the JCU we are referring to.

Table 2. I think there is a mistake in a covariate name “Percent forest version 2”; correct in final version. Why is a 50% CRI being reported? I think there is no need for that.

Authors’ comment: We´ve changed the name of the covariate. 

A 50% CRI is often reported to provide information on the magnitude and directionality of what is essentially half of the parameter estimate’s posterior density. Please see Petracca et al. (2019) in Animal Conservation for an example of 50% CRIs in the published literature. It is often helpful to use when the 95% CRI of a hyperparameter overlaps zero, but the 25th through 75th percentiles of the posterior density do not. It provides more nuance when it is often assumed that, just because a 95% CRI overlaps zero, there is no effect or directionality of that effect.

Line 356: Consider not citing work “in preparation”; not a real reference.

Authors’ comment: We´ve deleted this reference.

Lines 385-386: This is a speculation that needs some justification; whether jaguar or pumas establish between roads depends on how far apart roads are, what is between roads and along them.

Authors’ comment: We agree, we´ve changed the wording to better explain our point. 

Reviewer #2: The paper is an assessment of the mammal fauna in this area (which happens to have a corridor), coupled with a more in-depth analysis of the underlying drivers affecting the occupancy of jaguars and pumas in the region. Being in a corridor (or not) does not seem to be a contrast to be estimated in the model although the authors do compare some of the results between the three sub-study areas (one in a corridor and two not). As such the paper seems unfocused covering several questions with little rigor and no insights into the corridor issue itself.

Authors’ comment: Thanks for this input, and for your thoughtful review in general. We did indeed think about employing “corridor” as a categorical covariate in our modeling framework. However, we decided against this, hypothesizing that instead a continuous representation of these ideas (i.e., distance from strictly protected area and distance from Jaguar Conservation Unit at the grid level) would be more appropriate. Our motivation was our earlier work (Petracca et al. (2018)), which found “distance to strictly protected areas” to be an important covariate explaining jaguar habitat use within corridors in Central America. 

It is also important to consider that not all corridor areas are equivalent, with corridor units closer to JCU or protected area boundaries less likely to be degraded. It is also straightforward to derive estimates of richness by study area (corridor or not) post facto within a Bayesian framework, so we figured we would be able to investigate the corridor vs. non-corridor aspect without making “corridor” a categorical variable explicitly. This information may prove to be valuable to monitor species richness changes in time both in the corridor and in the adjacent areas (strictly protected areas or JCUs) that have different management types.

I have issues with the design of the study, in particular whether camera trap deployments were combined for a given cell or considered as independent measurement units within the cell. Also, camera traps were deliberately placed in trails, which can bias the measurement towards some species and increase the number of people captured. This is not ideal given that the assessment has a large geographical breadth and these placement decisions can affect the magnitude and precision of covariates at regional scales. The authors need to justify this design more clearly, explain how observations were aggregated for analysis, and explain what they mean by “independent observations”.

Authors’ comment: Data were combined from camera traps within each cell, as described in Lines 197-200. Additionally, about half of the cameras were placed on trails, but the other half was placed off trails as there are species that favor walking on trails but others avoid them, as you rightfully indicate. This was stated in Lines 190-192 and Figure 3. 

I was happy to see the use of modern statistical multi-species models being used in the analysis of these data and correcting for detection probability. However it was unclear how the authors arrived at their final model and how covariates were selected. I was disappointed that corridor/non-corridor was not a covariate in itself. The interpretation of some of the effects is incorrect (e.g. Fig 4) describing effects as positive or negative, when in reality they overlap with 0. The description of the statistical significance of some of the correlation coefficients is incorrect as well. When describing coefficients, effects or expected values the authors switch between mean+-confidence intervals and mean +- ? (Unclear wether it is standard error or standard deviation). For example when comparing the species richness between the three study areas (lines 312) it is unclear which parameter is used to describe uncertainty. If these are standard errors, there might not be any real difference between richness. If it is standard deviation, there might or might not be, the reader cannot assess it.

Authors’ comment: Thanks for this input, and for the praise that we are indeed employing a more modern statistical method (much appreciated!). 

In response to your comments, we acknowledge that we were not explicit enough in describing how we arrived at our final model (which was actually a maximum likelihood model selection strategy drawn from Petracca et al. (2018) in Journal of Applied Ecology). Reviewer 1 stated this as well. Hence, we have added material to the ms (L. 209-221) and have directed readers to the Supplementary Material for a full description of our approach. 

In addition, when one sees a 95% Bayesian credible interval that overlaps zero, a common reaction is to assume that the covariate has “zero” effect and that the effect is “non-significant.” The use of Bayesian statistics is a bit more nuanced, such that the overlap of a 95% BCI with zero does not mean there is no effect, but rather that there is imprecision in that estimate. We can indeed say that the parameter point estimate is positive or negative, but that there is imprecision. Don’t get me wrong, if a posterior density is centered directly on zero (please see our terrain ruggedness output for an example), we can be fairly confident that there is little to no effect. However, the 50% CRIs for two hyperparameters (percent forest cover and human presence) were above zero (and 90% of the posterior density for percent forest was above 0), suggesting that there is likely a directionality of that effect despite the imprecision (see Table 2; please also see Rich et a. 2016 in Journal of Applied Ecology for a highly-cited example of community-level hyperparameters having 95% BCIs that overlap zero). In general, having hyperparameters of low precision is common, especially if you are working with species with low detection probabilities and/or there is great variation in species-specific responses - luckily, we were able to see more precise estimates at the species level, using percent forest cover as an example in Figure 5.

As to our precision, for our Bayesian parameter estimates (i.e. the covariate betas) we provide the 95% CRI and 50% CRIs, which represent common summaries of posterior densities. You are correct that we failed to provide the unit of precision for our species richness estimates, which were derived parameters within the Bayesian model (rather than explicitly estimated parameters as the covariate parameters were). The unit is indeed SD here and we note that in the estimates provided. 

Both in the results and the discussion the authors seem to be more focused on the signs of the effects, than in their magnitude. Many of the effects discussed are not surprising (e.g. more forest more species). But there could be so much more here! In my opinion, the results —as presented— represent a lost opportunity since they might have important implications to better understand the effectiveness of corridors for mammals. As an example, if species richness within the corridor is not-significantly different compared to the more intact habitats being connected, this is an important result to be highlighted. Even if it is lower, the difference in mean species richness is only 1-2 species on average, which again is a very important result. These nuances in what the data is telling are key and often missed throughout the paper. The questions also seems too scattered, with some focusing on big cats, while others are looking at the whole community. It is confusing and difficult to get what the real results and messages of the research are.

Authors’ comment: Thanks for this feedback. Given that our model was based in Bayesian inference, we didn’t feel the need to report a frequentist result of whether that difference in species richness was indeed “significant” or “not significant.” However, we do agree that this difference in species richness is biologically not that meaningful and that we should have expanded upon this in our discussion. Please note our expanded content in Lines. 385-388, 416-421, where we note that while these numbers may appear similar, when we subset those values to large prey species only, there is a bigger difference between JCUs and corridor. We also added the number of independent records for large species in the discussion to further illustrate this difference. In addition, we are happy to do a frequentist test of significance if requested by the reviewer, though for now we still stay with the spirit of the Bayesian framework. We made changes to the results and discussion section to simplify our results and highlight what we consider are the takeaway point of the manuscript.

Authors, please do not be discouraged by these comments. I congratulate you for this great effort. My comments are made in a constructive spirit, to bring out what I think could be an exceptional paper. I believe this research should be published, but the field design needs to be clarified, the questions need to be tightened and more focused, the process to reach the final model explicitly stated and the interpretation of the results put into a wider context. I have attached the MS with some additional comments and suggestions that I marked directly. The paper should also be reviewed by an English copy editor before resubmission to improve the style according to journal standards. Hope all this is useful.

Authors’ comment: Thank you for your support of our work and for your careful review of our manuscript -- we do indeed hope we can see this through to publication! We have also gone through your additional comments and suggestions directly in the manuscript and have unilaterally incorporated these into the revised manuscript. Thank you again.

---

## [Decision Letter · Decision Letter 1]

16 Feb 2021

PONE-D-20-25172R1

Forest cover and occurrence of large-sized prey mediate jaguar (Panthera onca) and puma (Puma concolor) habitat use in a critical link of the Mesoamerican Biological Corridor

PLOS ONE

Dear Dr. Salom-Perez,

Thank you for submitting your manuscript to PLOS ONE. After careful consideration, we feel that it has merit but does not fully meet PLOS ONE’s publication criteria as it currently stands. Therefore, we invite you to submit a revised version of the manuscript that addresses the points raised during the review process.

I have now assesse your revision and received a second review from one of the two referees that reviewed your first submission. Both the reviewer and myself found your manuscript greatly improved, with previous criticisms properly addressed. However, the reviewer raises a point previsouly highlighted that I invite you to consider further, and adjust the text accordingly before your manuscript can be accepted.

We look forward to receiving your revised manuscript.

Kind regards,

Francesco Rovero, Ph.D.

Academic Editor

PLOS ONE

Reviewers' comments:

Reviewer's Responses to Questions

**Comments to the Author**

1. If the authors have adequately addressed your comments raised in a previous round of review and you feel that this manuscript is now acceptable for publication, you may indicate that here to bypass the “Comments to the Author” section, enter your conflict of interest statement in the “Confidential to Editor” section, and submit your "Accept" recommendation.

Reviewer #1: All comments have been addressed

2. Is the manuscript technically sound, and do the data support the conclusions?

Reviewer #1: Partly

3. Has the statistical analysis been performed appropriately and rigorously? 

Reviewer #1: Yes

4. Have the authors made all data underlying the findings in their manuscript fully available?

Reviewer #1: Yes

5. Is the manuscript presented in an intelligible fashion and written in standard English?

Reviewer #1: Yes

6. Review Comments to the Author

Reviewer #1: Authors have addressed implemented all changes or provided valid justification to most comments. However, I still find an important conceptual problem related to my first observation in the first round. Bellow I copy my original comment to authors, their reply, and a new comment based on their response and the changes in the manuscript.

Original reviewer comment: “Authors are hypothesizing that presence of puma and jaguar is associated with large prey species richness. However, in the Introduction I did not observe a rationale that supports that hypothesis. Arguably, prey biomass is more important than richness to determine predator presence. For example, in Los Llanos of Venezuela, 70% of jaguar diet is made up of 4 species (2 peccaries, capybara and giant ant eater) (Polisar et al 2003); a similar case is observed by Aranda (1994) in Mexico and other studies. It will be helpful to expand the introduction to address this hypothesis.”

Response by authors: “This is a great point by the reviewer, and we can certainly see the argument that prey biomass is a core component determining presence of predators. However, it has also been found that the richness of large-bodied prey can be a major driver of predator presence at large scales (please see Petracca et al. 2018 in Journal of Applied Ecology for an example including all jaguar corridors within Central America). It was the strong effect of this covariate in the Petracca et al. 2018 paper that led us to include this covariate in our manuscript - we have added material in our Introduction (L. 105-109) to help support our hypothesis that prey richness can be a viable driver of predator occupancy.”

I am not convinced by the response of the reviewers for the following reasons:

1. Petracca et al. (2018) are not evaluating the occurrence of jaguar as a function of prey diversity. That study, which is based on interviews (i.e, no estimate on game abundance/biomass), finds that there is an association between prey diversity and jaguar presence. Authors need to recognize that correlation does not imply causation. For example, prey diversity can be positively associated with forest cover and in general, a good conservation status. Better conserved sites likely have higher abundance of game, including large bodied species, which actually is the pattern found in the current study; lines 384–388 in Discussion state that “…it is clear that biomass of game is determining the presence of jaguar and puma, not game diversity.”

2. In the new version of the manuscript, while talking about jaguar and puma, authors state that (lines 106–109) “…their presence has been associated with prey biomass and availability [22,54–56]. However, recent studies have also found that prey richness can have an effect on large carnivore presence or ecology (e.g., diet) [15,57].” Here authors cite Petracca et al. 2018 (15), which as stated above does not prove that prey richness has an effect on large carnivore presence. Also, reference 57 (Ferretti et al. 2020) evaluates the effect of prey richness on dietary breath of large predators (i.e, increased generalism); it does not evaluate the presence of large predators as a function of prey diversity. Authors should be careful in how they present their interpretation of others’ studies.

3. To test the hypothesis that prey richness influences large carnivore presence, other research approach needs to be implemented. Confounding factors such as habitat quality and prey biomass need to be controlled.

In summary, I think the manuscript would be clearer and more robust without objective 3: “investigate the relationship between large carnivore habitat use and prey richness”. Currently the discussion feels forceful and somewhat contradictory because authors try to include that pattern (prey diversity as a cause of presence) as conclusive. However, that pattern is not sustained by the authors own findings and research design (i.e., higher biomass in sites with higher diversity; comparison of better with less conserved areas). Objectives 1 and 2 are clear and supported by authors methodology and results, and are sufficient to make an interesting and solid paper.

7. PLOS authors have the option to publish the peer review history of their article (what does this mean?). If published, this will include your full peer review and any attached files.

Reviewer #1: No

---

## [Author Response · Author response to Decision Letter 1]

8 Mar 2021

6. Review Comments to the Author

Reviewer #1: Authors have addressed implemented all changes or provided valid justification to most comments. However, I still find an important conceptual problem related to my first observation in the first round. Bellow I copy my original comment to authors, their reply, and a new comment based on their response and the changes in the manuscript.

Original reviewer comment: “Authors are hypothesizing that presence of puma and jaguar is associated with large prey species richness. However, in the Introduction I did not observe a rationale that supports that hypothesis. Arguably, prey biomass is more important than richness to determine predator presence. For example, in Los Llanos of Venezuela, 70% of jaguar diet is made up of 4 species (2 peccaries, capybara and giant ant eater) (Polisar et al 2003); a similar case is observed by Aranda (1994) in Mexico and other studies. It will be helpful to expand the introduction to address this hypothesis.”

Response by authors: “This is a great point by the reviewer, and we can certainly see the argument that prey biomass is a core component determining presence of predators. However, it has also been found that the richness of large-bodied prey can be a major driver of predator presence at large scales (please see Petracca et al. 2018 in Journal of Applied Ecology for an example including all jaguar corridors within Central America). It was the strong effect of this covariate in the Petracca et al. 2018 paper that led us to include this covariate in our manuscript - we have added material in our Introduction (L. 105-109) to help support our hypothesis that prey richness can be a viable driver of predator occupancy.”

I am not convinced by the response of the reviewers for the following reasons:

1. Petracca et al. (2018) are not evaluating the occurrence of jaguar as a function of prey diversity. That study, which is based on interviews (i.e, no estimate on game abundance/biomass), finds that there is an association between prey diversity and jaguar presence. Authors need to recognize that correlation does not imply causation. For example, prey diversity can be positively associated with forest cover and in general, a good conservation status. Better conserved sites likely have higher abundance of game, including large bodied species, which actually is the pattern found in the current study; lines 384–388 in Discussion state that “…it is clear that biomass of game is determining the presence of jaguar and puma, not game diversity.”

2. In the new version of the manuscript, while talking about jaguar and puma, authors state that (lines 106–109) “…their presence has been associated with prey biomass and availability [22,54–56]. However, recent studies have also found that prey richness can have an effect on large carnivore presence or ecology (e.g., diet) [15,57].” Here authors cite Petracca et al. 2018 (15), which as stated above does not prove that prey richness has an effect on large carnivore presence. Also, reference 57 (Ferretti et al. 2020) evaluates the effect of prey richness on dietary breath of large predators (i.e, increased generalism); it does not evaluate the presence of large predators as a function of prey diversity. Authors should be careful in how they present their interpretation of others’ studies.

3. To test the hypothesis that prey richness influences large carnivore presence, other research approach needs to be implemented. Confounding factors such as habitat quality and prey biomass need to be controlled.

In summary, I think the manuscript would be clearer and more robust without objective 3: “investigate the relationship between large carnivore habitat use and prey richness”. Currently the discussion feels forceful and somewhat contradictory because authors try to include that pattern (prey diversity as a cause of presence) as conclusive. However, that pattern is not sustained by the authors own findings and research design (i.e., higher biomass in sites with higher diversity; comparison of better with less conserved areas). Objectives 1 and 2 are clear and supported by authors methodology and results, and are sufficient to make an interesting and solid paper.

Authors´ response: Thank you very much for following up on this particular point in our manuscript. 

Given your insightful observations above, we acknowledge that the wording we used may have led readers to incorrectly interpret the correlation we found between prey species richness and puma/jaguar occupancy as causation. Our aim was not to prove that there was a causative link between prey richness and carnivore occupancy but rather to test for an association, with a particular interest in differences in the associations of large versus medium-sized prey richness on carnivore occupancy. Our emphasis on an association between carnivore and prey in several forms, rather than a causation, can be found in L. 105-110.

We also agree that the association between prey richness and carnivore occupancy should not be a formal objective and have removed Objective 3 from our manuscript (paragraph starting L. 111). We also eliminated any reference to a causative link between prey richness and carnivore occupancy (L. 449-464), and clarify that additional work will be needed to determine if prey biomass or forest cover (rather than richness) are the true causative agents. After careful consideration, we decided to leave the correlation results in the manuscript, as they have merit for what they are (i.e., correlations) while also serving as a discussion point for further research. 

Lastly, we have amended the title of our manuscript to focus on the strongest part of our manuscript (the community occupancy model): “Forest cover mediates large and medium-sized mammal occurrence in a critical link of the Mesoamerican Biological Corridor.”

---

## [Editor Report · Decision Letter 2]

11 Mar 2021

Forest cover mediates large and medium-sized mammal occurrence in a critical link of the Mesoamerican Biological Corridor

PONE-D-20-25172R2

Dear Dr. Salom-Perez,

We’re pleased to inform you that your manuscript has been judged scientifically suitable for publication and will be formally accepted for publication once it meets all outstanding technical requirements.

Kind regards,

Francesco Rovero, Ph.D.

Academic Editor

PLOS ONE

Additional Editor Comments (optional):

Thank you for the revised version and for thoroughly considering and addressing the remaining concern by the reviewer. I am pleased to now recommend acceptance of your manuscript.

---

## [Editor Report · Acceptance letter]

15 Mar 2021

PONE-D-20-25172R2 

Forest cover mediates large and medium-sized mammal occurrence in a critical link of the Mesoamerican Biological Corridor 

Dear Dr. Salom-Pérez:

I'm pleased to inform you that your manuscript has been deemed suitable for publication in PLOS ONE. Congratulations! Your manuscript is now with our production department. 

Kind regards, 

on behalf of

Dr. Francesco Rovero 

Academic Editor

PLOS ONE